# Nutritional Value of Microalgae and Cyanobacteria Produced with Batch and Continuous Cultivation: Potential Use as Feed Material in Poultry Nutrition

**DOI:** 10.3390/ani13213431

**Published:** 2023-11-06

**Authors:** Seyit Uguz, Arda Sozcu

**Affiliations:** 1Department of Biosystems Engineering, Faculty of Agriculture, Bursa Uludag University, 16059 Bursa, Turkey; 2Department of Biosystems Engineering, Faculty of Engineering and Architecture, Yozgat Bozok University, 66200 Yozgat, Turkey; 3Department of Animal Science, Faculty of Agriculture, Bursa Uludag University, 16059 Bursa, Turkey; ardasozcu@uludag.edu.tr

**Keywords:** *Scenedesmus* sp., *Ankistrodesmus* sp., *Synechococcaceae*, sustainability, protein

## Abstract

**Simple Summary:**

Alternative feed materials in poultry production are an important tool for sustainability and also improving animal performance, health status, and product quality. Thus, our present study aimed to investigate the usage possibility of algae as an alternative feed ingredient in poultry nutrition. In these regards, this study aimed to evaluate the nutritional value and production cost of three species of algae, namely *Scenedesmus* sp., *Ankistrodesmus* sp., and *Synechococcaceae*, harvested with batch and continuous cultivation processes. The current results clearly showed that the nutritional composition and amino acid profile of algae biomass harvested from three different microalgae species under batch and continuous cultivation practices are relatively higher and show variations in protein and lipid content. The current findings indicate the superiority of *Scenedesmus* sp. for its high PUFA and lysine content, and *Synechocccaceae* for its high content of methionine and threonine. Furthermore, *Synechococcaceae* could be offered as a natural additive for the pigmentation of egg yolk and broiler meat due to its darker shade of yellowness. It is important to emphasize that the nutritional value and production cost of microalgae must be considered to choose the right one as an alternative feedstuff.

**Abstract:**

Recently, the demand for new alternative feedstuffs that do not contain chemical residue and are not genetically modified has been increased for sustainability in poultry production. In this respect, the usage of algae as animal feed is very promising as an alternative feed ingredient that reduces pollutant gases from animal production facilities. The aim of the current study is to investigate the usage possibility of algae, through determining nutritional value and production cost, as a feed ingredient in poultry nutrition. Three microalgae species, including *Scenedesmus* sp., *Ankistrodesmus* sp., and *Synechococcaceae*, were produced with batch and continuous cultivation to determine the difference in the lipid, protein, carbohydrate, fatty acid, and amino acid profiles, as well as the color characteristics and production cost. The highest lipid content of 72.5% was observed in algae biomass produced from *Synechococcaceae* with batch cultivation, whereas the highest protein level was found in algae biomass produced by *Synechococcaceae* under continuous cultivation practice (25.6%). The highest content of PUFA was observed in *Scenedesmus* sp. harvested from both batch and continuous cultivation (35.6 and 36.2%), whereas the lowest content of PUFA was found in *Synechococcaceae* harvested with continuous cultivation (0.4%). Continuously cultivated of *Scenedesmus sp*. had higher carbohydrate content than batch-cultivated *Scenedesmus* sp. (57.2% vs. 50.1%). The algae biomass produced from *Synechococcaceae* was found to have a higher content of essential amino acids, except lysine and histidine, compared to *Scenedesmus* sp. and *Ankistrodesmus* sp. Cultivation practices also affected the amino acid level in each algae species. The continuous cultivation practice resulted in a higher level of essential amino acids, except glycine. *Synechococcaceae* had richer essential amino acid content except for proline and ornithine, whereas continuous cultivation caused an incremental increase in non-essential amino acids. The lightness value was found to be the lowest (13.9) in *Scenedesmus* sp. that was continuously cultivated. The current study indicated that *Scenedesmus* sp. could be offered for its high PUFA and lysine content, whereas *Synechococcaceae* could have potential due to its high content of methionine and threonine, among the investigated microalgae and Cyanobacteria.

## 1. Introduction

Poultry products have a significant role in human nutrition due to their high-quality proteins, lower lipid content, and their lower cost, especially in developing countries [1]. Consumer demand shows a significant increase due to growth in the human population. Also, consumer preference have shown some changes away from conventional poultry production systems to natural and sustainable production systems, including free-range and organic egg and broiler meat production [2]. Therefore, a demand for new feed materials has been increasingly gaining importance in sustainable production. New alternative feed sources that do not contain chemical residue and are not genetically modified (GMO) are needed replace GMO maize, soybean meal, and animal protein sources, especially in organic poultry production [3]. It should be emphasized that algae can be used to mitigate pollutant gases from animal production facilities through giving support to sustainable poultry production [4].

Recently, algae have been studied for use as human supplements and as feed material for livestock production. Algae’s nutritional profile includes carbohydrates, essential fatty acids and amino acids, carotenoids, and vitamins A, B1, B12, C, D, and E [5,6,7,8]. Algae are classified into two groups, macro-algae and micro-algae. Both macro-algae (for example, *Laminaria*, *Gracilaria*, *Ulva*, *Padina*, and *Pavonica*) and micro-algae (*Chlorella*, *Tetraselmis*, *Spirulina*, *Nannochloropsis*, *Nitzchia*, *Navicula*, *Chaetoceros*, *Scenedesmus*, *Haematococcus*, and *Crypthecodinium*) can be used as feed for farm animal nutrition [9].

The usage of algae as animal feed could be beneficial in improving animal health, performance, and product quality. Current studies clearly demonstrated that adding of micro-algae (e.g., *Chlorella*, *Scenedesmus*, and *Arthrospira*) into traditional diets has beneficial effects on animal growth, health status, and physiological processes, as well as improving the quality and quantity of meat and eggs [10]. The beneficial effects were lower cholesterol and improved immunity [11], animal growth and improved meat quality [12], increased reproductive performance [13], antiviral and antibacterial action offering strong resistance to diseases [14], improved gut function and the colonization of probiotics in the intestinal tract [15,16], and an enhancement in feed conversion efficiency [17]. Nowadays, approximately 30% of the world’s production of microalgal biomass is for animal nutrition [18,19]. The use of microalgae as a feed supplement is currently being practiced mainly in the United States and the United Kingdom [16,20]. The production is spreading to many Asian countries, including Japan, the Philippines, China, and Korea [21]. However, there are issues in producing microalgae biomass as feed in a sustainable and economical feasible way [8].

Another critical issue is the nutritional quality of algae, which can be varied by algae species, chemical composition, and growth conditions. The cultivation mode, whether batch or continuous, has a significant impact on the nutritional value of algae. Continuous cultivation allows for better control of the growth rate and metabolic activity, leading to higher production of valuable compounds such as proteins, fatty acids, vitamins, and minerals [22]. To evaluate the effect of the algae species and cultivation modes (batch and continuous) on the nutritional composition and amino acids profile for algae usage as a feed ingredient in poultry rations, different microalgae species, including *Scenedesmus* sp. (AQUAMEB-60), *Ankistrodesmus* sp. (AQUAMEB-33), and *Synechococcaceae* (AQUAMEB-32), were produced with batch and continuous cultivation practices.

## 2. Materials and Methods

### 2.1. Algae Cultivation and Medium

Two microalgae species (*Ankistrodesmus* sp. (AQUAMEB-33) and *Synechococcaceae* (AQUAMEB-32)) used in this study were isolated from Lake Uluabat, Turkey. *Scenedesmus* sp. (AQUAMEB-60) was isolated from a freshwater aquarium. All microalgal species were obtained from AQUAMEB Culture Collection of Algae and Cyanobacteria (Aquatic Microbial Ecology and Biotechnology Laboratory, Bursa, Turkey). These strains were selected due to their high adaption to a wide range of environmental conditions and high growth rate. Each strain was inoculated in 250 mL Erlenmeyer flasks containing 100 mL of Bold’s Basal medium (BBM) [4,23] that consisted of K_2_HPO_4_ (175 mg L^−1^), CaCl_2_·2H_2_O (25 mg L^−1^), MgSO_4_·7H_2_O (75 mg L^−1^), NaNO_3_ (250 mg L^−1^), K_2_HPO_4_ (75 mg L^−1^), MoO_3_ (1.42 mg L^−1^), NaCl (25 mg L^−1^), EDTA (50 mg L^−1^), KOH (31 mg L^−1^), FeSO_4_·7H_2_O (4.98 mg L^−1^), H_2_SO_4_ (0.1 µL L^−1^), H_3_BO_3_ (11.42 mg L^−1^), and a trace metal solution (1 mL L^−1^). The trace metal solution was composed of ZnSO_4_·7H_2_O (8.82 g L^−1^), MnCl_2_·4H_2_O (1.44 g L^−1^), CuSO_4_·5H_2_O (1.57 g L^−1^), and CO(NO_3_)_2_·6H_2_O (0.49 g L^−1^). The medium pH was adjusted to 6.8–7.0 with 0.5 M HCl or 0.5 M NaOH, and the prepared medium was sterilized via placing it into an autoclave for 20 min at 121 °C before use. The Erlenmeyer flasks were placed in front of two daylight LED lamps (ACK Lighting, Istanbul, Turkey).

### 2.2. Experimental Procedure

To test the effect of growing conditions (batch and continuous culture) and algae species on the amino acid and nutrient content of the algae, algae cultures were periodically doubled and then transferred to 10 L flat plate PBRs (constructed as published by Uguz et al. [4], Figure 1) upon reaching a volume of 10 L (Table 1). During the cultivation, the algal culture was continuously aerated with air (containing 1200 ppm CO_2_) at a rate of 0.5 L min^−1^ per liter of culture volume. A flow meter (Bass Instruments, Istanbul, Turkey) was used to maintain the gas volumetric flow rate. Each species was grown in triplicate with control tanks at a light intensity of 200 μE m^−2^ s^−1^ (24 h light photoperiod). Towards the end of each test, the algal culture in the control PBRs was taken as inoculum for the next experiment. The light intensity was measured with a LI-COR quantum sensor (LI-COR, Lincoln, Nebraska). Environmental conditions, including pH, temperature, water level (maintained with deionized water), aeration rate, and light intensity, were monitored and controlled throughout the experiments. pH was adjusted daily to maintain a range of 6.8–7.00 through adding 0.5 M HCl or 0.5 M NaOH as needed. Room temperature was maintained between 23 and 25 °C during the 7-day experimental period. The temperature and pH of the algal culture were measured using a digital pH meter (Hanna Instrument, Ankara, Turkey, HI98128). The algae were grown under these conditions for seven days, and cell counts were measured daily using hemocytometers under an optical microscope. There was no nutrient addition in the batch culture experiments from the beginning to the end of the cultivation period.

The dilution rate of 0.1 [24] was applied to the PBRs for the continuous cultivation culture experiments. Ten percent of the total working volume (10 L) was removed daily from each PBR, and the same amount of BBM was added back to each PBR.

### 2.3. Analytical Methods

A total of 5 samples of harvested dry algal biomass were analyzed to determine the chemical composition according to the procedures of the Association of Official Analytical Chemists (AOAC, 2006). Lipid content was analyzed with acid hydrolysis (AOAC, Official Method 954.02, 2006). According to this method, the samples were hydrolyzed with hydrochloric acid and then transferred to a mojonnier flask. Hexane and ethyl ether were added for further fat extraction. The extracts were dried, reconstituted in hexane, and filtered via a column of sodium sulfate. These filtered extracts were evaporated, dried, and finally weighed. The crude protein was determined using the Kjeldahl method, using 6.25 as a conversion factor to calculate protein content (AOAC, Official Method 984.13 (A–D), 2006). Carbohydrate composition was determined from their alditol acetates via gas–liquid chromatography (GLC) and via gas–liquid chromatography–mass spectrometry (GLC–MS) based on the procedure by Oxley et al. [25].

### 2.4. Determination of Fatty Acids and Amino Acids

In this study, the lyophilized microalgae biomass samples were analyzed via gas chromatography to quantify the fatty acid methyl esters (FAMEs). A gas chromatograph (Agilent 6890N Series) was used to analyze the FAMEs and was equipped with a flame ionization detector (Agilent Technologies Inc., Wilmington, DE, USA). The column temperature was 120 °C for 1 min, then heated thermally to 175 °C at a rate of 10 °C/min, maintained for 10 min at 175 °C, heated again from 175 to 210 °C at a rate of 3 °C/min, maintained at 210 °C for 5 min, heated further from 210 to 240 °C at a rate of 5 °C/min, and maintained at 240 °C for 5 min. An HP 88-MS capillary column (100 m × 0.25 mm i.d., 0.20 µm film thickness; Agilent Technologies Inc.) was used for the analyses, with helium as the carrier gas flowing at 2 mL/min. One microliter of FAMEs was injected into the helium gas. The injector and flame ionization detector were maintained at 250 and 280 °C, respectively. The quantity and composition of FAMEs in the samples were determined from peak areas of identified FAMEs and the internal standard (Mixture ME-100; Greyhound Chromatography and Allied Chemicals, Birkenhead, Merseyside, UK).

The microalgal biomass was harvested for amino acid profile analysis at the end of cultivation (day 7). The pellet was freeze-dried and stored at −40 °C in a lyophilizer for future experiments. The amino acid LC–MS/MS analysis was performed with an Agilent LC–MS mass spectrometer (Agilent Technologies, Santa Clara, CA, USA). The targeted amino acid concentrations were determined using electrospray ionization (ESI) and multiple reactions monitoring (MRM).

A total of 0.5 g of the dry algal biomass sample was hydrolyzed with 4 mL of acidic hydrolysis reagent in a screw-cap tube for 24 h at 110 °C. The hydrolyzed samples were centrifuged (4000 rpm, 5 min, Hettich Universal 320 desktop air-cooled centrifuge). Afterwards, 10 µL of supernatant was transferred to the sample bottle and diluted 800 times with distilled water. The HPLC system injected 3 μL of prepared biomass sample into the Jasem analytical column specified for amino acid analysis, which was maintained at 30 °C. The chromatographic separation was conducted using Jasem’s mobile phases A and B with gradient elution at a flow rate of 0.7 mL/min [26].

The HPLC elution process was started through increasing the A gradient from 22 to 78% in 3 min, held at 78% for 0.5 min, continued through decreasing from 78 to 22%, and then equilibrated at 22% A for 3 min. Mass spectra (MS) were measured in the positive ion mode of the electrospray ionization (ESI) source. The optimal MS detector settings were as follows: drying gas temperature was 150 °C, and capillary voltage was 2000 V (+). Full scan mass spectra mode was operated for the amino acid detection and IS over an *m*/*z* range of 30–3000. Multiple-reactions monitoring (MRM) of the amino acid and IS were carried out at optimum fragmentation voltages (FV) and optimum collision energies (CEs). The peak area ratio of the amino acid to the assigned IS was evaluated to quantify the targeted amino acid concentration [27].

### 2.5. Determination of Color Characteristics

The color characteristics as lightness (L*), redness (a*), yellowness (b*), chroma (C), and hue value (α) of harvested dried algal biomass (*n* = 5 samples per each experimental group) were measured using a spectrophotometer (Konica- Minolta, Osaka, Japan).

### 2.6. Determination of Growth Kinetic Parameters

The dry biomass measurements were conducted every 24 h. The dry biomass concentration was determined using a 25 mL sample of algal culture, which was subsequently collected and filtered using a 0.45 µm pre-dried membrane filter paper. The algae-laden filter was then dried at 105 °C for 3 h in a lab oven and weighed using an analytical balance. The concentration of algal biomass was calculated using Equation (1) [4]:(1)$$Dry\;algal\;biomass\;conc.\;\left(gL^{-1}\right)=\frac{Final\;weight\;\left(g\right)-Initial\;weight\;(g)}{Sample\;volume\;\left(mL\right)\times 1(L)}\times 1000\;(mL)$$

The biomass productivity (*P*, mgL^−1^ day^−1^) was calculated according to Equation (2) [28]:(2)$$P_{biomass}=∆x/∆t$$
where Δx is the dry biomass concentration (gL^−1^) within a cultivation time of Δt (day). The productivity of carbohydrates, lipids, and proteins was calculated using Equation (3) [29]:(3)$$P_{lipids,\;carbohydrate,\;protein}(mg\;L^{-1}{day}^{-1})=P_{biomass}\times C_{f}$$
where *P*_lipids,carbohydrate,protein_ is the productivity of carbohydrates, lipids, and proteins (mgL^−1^ day^−1^); *P_biomass_* is the biomass productivity (mgL^−1^ day^−1^); *C_f_* is the final content of lipids, carbohydrates, and proteins given as percent dry weight.

### 2.7. Estimating Energy Used

The energy used for pumping and lighting was calculated using the electric power consumption of air pumps and LED lamps. Electric power consumption was calculated (https://www.rapidtables.com, accessed on 24 October 2023).
E _(kwh/day)_ = P_(W)_ × t _(h/day)_/1000 _(W/kW)_(4)
where *E* is the power consumption in kilowatt-hours (kWh) per day, P(w) is the rated power of apparatus in watts (W), and t is the time duration in the number of usage hours per day. The power consumption was translated into dollars (1 kWh costs 12.7 cents in Bursa, Turkey) to calculate the cost (USD).

### 2.8. Statistical Analyses

The collected data were analyzed with to one-way Analysis of Variance (ANOVA) using JMP (version 13) followed by the least significant difference (LSD) Student comparison tests to compare the differences between treatments, where significant differences were observed among the treatments. Results are given as the mean and standard error of the mean (SEM). Analysis for percentage data was conducted after an arcsine transformation of the data. Pearson correlation was used to determine the correlation between protein content and some amino acid levels in algae biomass. Differences were considered significant at *p* ≤ 0.05.

## 3. Results and Discussions

During recent years, algae sources are generally considered as a rich source of proteins, amino acids, lipids, n-3 long-chain polyunsaturated fatty acids, polysaccharides, minerals, vitamins (especially water-soluble vitamins), enzymes, and pigments, such as chlorophyll, antioxidants, and carotenoids [30,31,32]. The current results clearly showed that both the nutritional composition and amino acid profile of algae biomass harvested from three different microalgae species under batch and continuous cultivation practice are increased. The nutritional content of microalgae is relatively higher and shows variations in protein and lipid contents compared to the other feed materials in poultry nutrition [3]. The nutritional content (protein, carbohydrate, lipids, vitamins, and minerals) depends on the microalgae species and growing conditions [12]. Microalgae used in animal feed have been investigated and shown to have high nutritional value for poultry, pigs, cows, and aquaculture [33,34].

Table 2 shows the biomass production and biochemical composition of *Scenedesmus* sp., *Ankistrodesmus* sp., and *Synechococcaceae* under batch and continuous operations. *Scendesmus* sp. was identified as a fast-growing strain with the highest biomass productivity. For *Ankistrodesmus* sp. and *Synechococcaceae*, biomass productivity of 114.2–164.2 mg L^−1^ day^−1^ and 71.4–119 mg L^−1^ day^−1^ were calculated, respectively.

Significant interactions of algae species and cultivation modes were observed for the lipid, protein, and carbohydrate contents of algae biomass (Table 2). The highest lipid content of 72.5% was observed in algae biomass produced from *Synechococcaceae* under batch cultivation (*p* < 0.0001). Based on high lipid content, *Synechococcaceae* was identified as the strain with the highest lipid productivity of 86.2 mg L^−1^ day^−1^ (Table 2). It has been reported that the lipid content of algae biomass produced from different species shows changes from 1% to 40%; however, in some cases, it could increase to 85% of the biomass on a dry matter basis [35]. Hempel et al. [29] achieved the lipid productivity range between 72.5 and 121 mg L^−1^ day^−1^ for the *Chlorella* sp. strain.

The protein content of algae biomass was affected by both algae species and cultivation practices (*p* < 0.0001). The highest protein level was observed in algae biomass produced by *Synechococcaceae* harvested from the continuous cultivation method (25.6%). Unlike current results, it was reported that algae biomass harvested from several algae species (*Anabena cylindrical*, *Spirulina*, *Chlorella vulgaris*, and *Synechococcus* sp.) had a crude protein content between 43 and 73% [36]. Previous reports clearly demonstrated that some microalgae, such as *Spirulina* and *Chlorella*, could be accepted as a good protein source with favorable amino acid composition [37], which has a similar protein and fatty acid composition to animal protein used in poultry feed [10,31]. The current findings clearly show that the algae biomass has a comparable protein level to conventional feed ingredients, such as soybeans.

Determining the volumetric productivity of protein allows for the prediction of synthesized proteins, both per day and per liter. *Synechococcaceae* has the highest protein level (25.6%) compared to other strains. However, *Scendesmus* sp. was identified as the top protein producer based on high biomass productivity. As a result of screening experiments, three algae species have higher protein contents under continuous cultivation mode, but the protein productivities were higher under batch cultivation modes because of the higher biomass productivities. This shows that the main influencing factor for protein productivity is biomass productivity, and this phenomenon shows similarity to those reported by Hempel et al., (2012) [29] for amino acid productivity.

In animal nutrition, the most widely used microalgae for protein-rich feed supplements are *Chlorella*, *Arthrospira*, *Dunaliella*, *Tetraselmis*, *Phaeodactylum*, *Skeletonema*, and *Scenedesmus* [38,39]. It is estimated that microalgae could produce approximately 2.5 to 7.5 tons of protein per hectare in each year [7,40]. Therefore, this source has the potential to be a candidate as an alternative and sustainable protein source. However, it should be considered that there are some factors affecting the protein content of algal biomass, such as the culture conditions and the species. For example, microalgae that are grown heterotrophically have a higher content of protein compared to the microalgae grown autotrophically. This difference causes a reduction in photosynthetic pigments, mainly nitrogen-rich chlorophyll, and subsequently a downgrade for the metabolism of protein production, because autotrophic algae's exposure to sunlight is limited due to shading. This is a critical fact due to its relation to cultivation conditions. The lower protein content in this study may also be explained by the autotrophic growing condition.

The carbohydrate content showed significant differences between algae species and cultivation practice. Interestingly, a significant difference was observed between *Scenedesmus* sp. and the other two species, *Ankistrodesmus* sp. and *Synechococcaceae* (53.7% vs. 14.7% and 7.1%, respectively, *p* < 0.0001). Continuously cultivated *Scenedesmus* sp. had a higher carbohydrate content than the batch-harvested *Scenedesmus* sp. (57.2% vs. 50.1%, *p* < 0.0001). These differences observed among the algae biomasses could be attributed to microalgae species. According to a previous review adapted by Christaki et al. [41], the percentage of carbohydrates on the basis of dry matter was found to be 12–17% for *Chlorella vulgaris*, 8–14% for *Spirulina maxima*, 10–17% for *Isochrysis galbana*, and 32% for *Diacronema vikianum*. Carbohydrates are an important component of algae biomass due to their nutritional and pharmaceutical value. For example, beta-1-3-glucan is of soluble fiber and mainly found in *Chlorella* sp., and it has an antioxidant effect for reducing blood cholesterol levels [38,39,42].

Stress conditions and nutritional deficits can both be implemented during the cultivation of microalgae to promote lipid and protein accumulation. The induction of lipid synthesis through stress conditions, such as nitrogen deprivation or oxidative stress, can lead to increased lipid content in microalgae cells [43,44]. Additionally, the manipulation of nutritional factors, such as nitrogen and phosphorus levels, can influence microalgal growth rates, biomass yield, and the nutritional content of lipids and fatty acids [45]. The implementation of stress phase and nutritional deficit strategies can be effective in enhancing lipid and protein accumulation in microalgae cultivation. Kilham et al. [46] investigated the effect of nutrition limitations on the biochemical composition of *Ankistrodesmus falcatus*. They reported that nutrient limitation could affect the biochemical constituents of algae, with low-nitrogen cells having lower protein-to-lipid ratios and higher carbohydrate content [46]. Environmental factors and nutrient availability also influence the amount of carbon fixed in lipids and carbohydrates in algae [47]. Another research work using the *Cyanobacterium spirulina platensis* found that increased CO_2_ concentrations increased the amount of carbohydrates in the cells while reducing the relative concentrations of proteins and pigments [48]. Different species of algae have varying carbohydrate and protein contents, with brown algae generally having higher carbohydrate content and lower protein content [49]. Overall, nutrient availability, environmental factors, and species-specific characteristics can influence algae’s protein and carbohydrate contents. In the current study, the highest level of lipids and proteins with the lowest level of carbohydrates is highlighted. This could be related to the nutrient accumulation characteristics of the algae *Scenedesmus* sp. and *Ankistrodesmus* sp., which accumulated less carbohydrates than *Synechococcaceae*, and both algae accumulated photosynthetic products in the form of carbohydrates (53.7% and 14.7% (*Scendesmus* sp.) vs. 7.1% (*Ankistrodesmus* sp.)). Similar results were also reported by Brown [50].

A comparison of saturated fatty acids (SFA) and unsaturated fatty acids (monounsaturated fatty acids, MUFAs, and polyunsaturated fatty acids, PUFAs) of different microalgae species (the *Scenedesmus* sp., *Ankistrodesmus* sp., *Synechococcaceae*) and cultivation practices is shown in Figure 2. Significant variations were observed for fatty acids by microalgae species and cultivation mode. The percent ratio of SFA in the total fatty acids was found to be the highest with a value of 72.5% in algal biomass obtained from *Synechococcaceae* harvested from batch cultivation (*p* < 0.0001). The MUFA content was observed to be the highest in *Ankistrodesmus* sp. harvested from batch cultivation (45.7%, *p* < 0.0001) and ranged between 22.3% and 40.8% in other microalgae species and for the continuous cultivation practice. On the other hand, the PUFA content of algal biomass changed from 0.4% to 36.2% (*p* < 0.0001). The highest content of PUFAs was observed in *Scenedesmus* sp. harvested from both batch and continuous cultivation, whereas the lowest content of PUFA was found in *Synechococcaceae* harvested from the continuous cultivation process. As previously mentioned, the fatty acid content of microalgae could show differences due to the taxonomic group and growth conditions [50,51]. Indeed, the current results clearly show that the cultivation processes of batch and continuous cultivation resulted in significant differences between the algae species for the composition of fatty acids.

It has been reported that feed sources with a rich content of PUFAs have preventive effects against some diseases, due to their significant roles in synthesizing hormones that regulate blood clotting, arterial wall contraction and relaxation, and supporting immune function, brain development, visual function, etc. [52]. These fatty acids support the health status of humans and animals due to their contribution to the production of prostaglandins, thromboxane, etc., which are biologically active substances, and have importance in reducing cholesterol and triglycerides in the blood. Bird et al. [53], Simopoulos [54], and Gouveia et al. [9] clearly demonstrated that PUFAs have preventive effects against cardiovascular diseases, atherosclerosis, skin diseases, autoimmune diseases, and some forms of cancer. Many species of microalgae have a significant content of PUFAs, mainly eicosapentaenoic acid (EPA) and docosahexaenoic acid (DHA) [55], that could not be synthesized by humans or animals [54].

According to previous studies, microalgae supplementation could be made appropriately in poultry nutrition, as a partial replacement for soybean meal, or as a fat source in diets [56,57]. According to Petrolli et al. [58], the supplementation of microalgae produced by *Schizochytrium* sp. stimulated growth and caused an enrichment in DHA content in the thigh and drumstick in broilers. The increase in body weight could be explained by the stimulant effect of DHA on the development of nerve tissue in the early phase of body development. In another study performed by Keegan et al. [56], the broilers’ diet supplemented with *Aurantiochytrium limacinum* algae resulted in an increase in the DHA content of breast and thigh meat and the total content of ω-3 PUFA but reduced the ω-6/ω-3 ratio of the meat. These changes are accepted as desirable for improving meat quality through a safe and sustainable method for human consumption. Furthermore, in laying hen nutrition, dietary supplementation with microalgae *Nannochloropsis* sp. increased the ω-3 PUFA content of egg yolk [57].

The essential and non-essential amino acid composition (mg/100 mg of dried weight) is illustrated in Table 3 and Table 4. The algae biomass produced from *Synechococcaceae* was found to have a higher content of essential amino acids, except lysine and histidine, compared to *Scenedesmus* sp. and *Ankistrodesmus* sp. (*p* < 0.001). The most abundant amino acids were leucine (1.10 mg/100 g), lysine (0.93 mg/100 g), and threonine (0.79 mg/100 g) in *Scenedesmus* sp. The most abundant amino acids in *Ankistrodesmus* sp. were leucine (1.24 mg/100 g), lysine (0.88 mg/100 g), and Arginine (0.6 mg/100 g). In *Synechococcaceae,* the most abundant amino acids were leucine (1.38 mg/100 g), arginine (0.97 mg/100 g), and lysine (0.85 mg/100 g). In each algae species, the amino acid content was also affected by cultivation practice. Continuous cultivation practice caused a higher content of essential amino acids, except glycine (*p* < 0.001). Similar results were observed for non-essential amino acids. *Synechococcaceae* had a richer content of non-essential amino acids except for proline and ornithine. The continuous cultivation practice caused an increment in non-essential amino acids (*p* < 0.001).

Most microalgae contain essential amino acids which cannot be synthesized by humans and animals, unlike other feed materials from plants [9]. A previous study reported that the content of amino acids—lysine, methionine, tryptophan, threonine, valine, histidine, and isoleucine—for some microalgae species is comparable to that of egg or soy [18]. When compared to the amino acid profile of soybeans, the amount of essential amino acids such as lysine, cysteine, and tryptophan are lower, whereas other essential amino acids, including methionine, threonine, and isoleucine, are comparable or higher [35]. Significant relationships between amino acids and protein content in all algae species are presented in Figure 3. As seen in Figure 3, the protein content showed an increase in the content of arginine, threonine, valine, isoleucine, leucine, methionine, lysine, phenylalanine, and histidine in *Ankistrodesmus* sp. and *Synechococcaceae*, and *Scenedesmus* sp. *(p* < 0.001).

Regarding the rich content of protein and amino acid profiles of algae biomass, microalgae could be accepted as an alternative to traditional protein sources in poultry nutrition. Microalgae produce protein and can be a potential renewable source of protein in animal feed [59,60]. According to an FAO/WHO report, *Spirulina* can be used as a feed for domestic animals. Microalgae, e.g., *Chlorella* and *Arthrospira* (*Spirulina*), are considered sustainable sources of protein due to their essential amino acid profile, which is like conventional protein sources such as soybeans and eggs [61].

When regarding the protein productivity value (in Table 2), there is an inverse ratio between protein content and productivity of algal biomass. That is, while the protein content shows an increment, the value of protein productivity declines in all species of microalgae. As seen in Figure 3, according to the protein content, the alteration pattern of amino acids differs with soft and sharp inclines in microalgae species. For example, methionine and lysine are limiting amino acids in poultry nutrition. The correlation between the methionine and protein content of *Synechococcaceae* is a sharp and strong correlation compared to the other species. On the other hand, the correlation between lysine and protein content was sharper in both *Scenedesmus* sp. and *Synechococcaceae.* Due to the differences between the amino acid profiles and the first limiting amino acids of pulses and microalgae, the ingredients could be combined to achieve a balanced amino acid profile [62].

Egg yolk color is an important quality and visual parameter for consumers, and the required degree of pigmentation changes from golden to orange–yellow colors, which are considered more attractive colors for consumers [63]. Due to inability of laying hens to synthesize the carotenoids, which has the effect of intensifying the egg yolk color, laying hens’ diets need to be supplemented with some ingredients with a rich content of carotenoids.

In nature, carotenoids are synthesized by algae, fungi, some plants, and bacteria [64]. The color evaluation of algal biomass was estimated through the colorimetric determination of lightness (*L**), redness (*a**), and yellowness (*b**) indexes. The color characteristics of algae biomass from different microalgae species and cultivation practices are presented in Table 5. The lightness (*L**) value was found to be the lowest, with a value of 13.9 in algae biomass produced from *Scenedesmus* sp. that was continuously cultivated (*p* < 0.001). An *L** value nearer to zero means that the color characteristic of the measured material is a darker color, such as black. The *a** value is used for determining the red and green characteristics, and *b** is an indicator of the yellow and blue characteristics. In this study, according to *a** and *b** values, the color characteristics of algae samples were found to be in the range of green and blue colors. These color characteristics have importance indicating the pigmentation of egg yolk and meat. Previous studies focused on pigmentation demonstrated the beneficial effects of microalgae supplementation in broilers’ and layer hens’ diets. Toyomizu et al. [65] reported that when the broiler diet was supplemented with algae biomass of *Arthrospira platensis* (4 and 8% supplementation level), the pigmentation of the meat (yellowness and redness) increased. Similar results were also observed for broilers when feed was supplemented with *Arthrospira platensis* [66]. While Wu et al. [57] demonstrated that when the diet is supplemented with microalgae *Nannochloropsis* sp. biomass, an increase in yolk color score was obtained. The score indicated that the yolk was a darker yellowish color.

Table 6 shows the economic estimates of the PBR system. This lab-scale study used 10 L PBRs to test the effect of batch and continuous cultivation modes and algae species on the amino acid and nutrient composition of the algae. The estimated capital cost of this lab-scale system was USD 519.24, including flowmeters, pumps, and PBR tanks (Table 6). A 1.5 hp air pump was used to transfer the air to each PBR. Power consumption for lighting and pumping is the only energy used by the PBR system. Pumping (and aeration) cost ~USD 0.034 L^−1^ day^−1^, and lighting cost ~USD 0.0024 L^−1^ day^−1^ during the experiment, based on the electricity rate in Bursa, Turkey. The medium preparation and refilling cost USD 0.02 and USD 0.032 L^−1^ day^−1^ for batch and continuous cultivation modes, respectively.

Algal biomass and protein production costs under batch and continuous cultivation modes for *Scenedesmus* sp., *Ankistrodesmus* sp., and *Synechococcaceae* are shown in Figure 4. With the current PBR system, the total operating cost for producing 1.0 g L^−1^ d^−1^ dry algal biomass was USD 0.24–0.37 L^−1^ d^−1^, USD 0.34–0.6 L^−1^ d^−1^, and USD 0.47–0.95 L^−1^ d^−1^ for *Scendesmus* sp., *Ankistrodesmus* sp., and *Synechococcaceae*, respectively. While the lowest price of dry algal biomass was achieved with *Scenedesmus* sp., the highest price was for *Synchococcaceae.* The total operating cost for producing 1.0 g L^−1^ d^−1^ protein was USD 1.6–2.1 L^−1^ d^−1^, USD 2.0–2.7 L^−1^ d^−1^, and USD 2.3–3.7 L^−1^ d^−1^ for *Scendesmus* sp., *Ankistrodesmus* sp., and *Synechococcaceae*, respectively.

These are much higher costs compared to the other feed materials in poultry nutrition. It is notable that the cost estimates were derived for 10 L PBRs. It is worth mentioning that the cost estimates were obtained for 10 L photobioreactors (PBRs). According to Acien et al. [67], scaling up the production capacity by 2.2 times resulted in an 82% reduction in the algae production cost. Therefore, scaling up the PBR (photobioreactor) system promises to decrease operational expenses significantly. The produced algae have potential to reduce the overall cost when it is used or sold as a valuable product.

The economic analysis shows a need to enhance the design of aeration and lighting systems. The provision of air supply, mixing, and illumination should be designed to meet the specific requirements of algal biomass production. The effective use of natural lighting can also lead to a reduction in expenses related to artificial lighting.

## 4. Conclusions

The interest in microalgae as a food and feed raw material is expected to increase significantly due to the demand for plant-based products and the healthy nutritional value of microalgae. It could be concluded that the feed material produced by microalgae could be efficiently used due to its chemical composition to enhance the nutritional composition of meat and eggs through the partial replacement of conventional dietary protein sources. We investigated the nutritional value and production cost of *Scenedesmus* sp., *Ankistrodesmus* sp., and *Synechococcaceae*, produced with batch and continuous cultivation. The current findings indicate that *Scenedesmus* sp. has a superiority for its high PUFA and lysine content, and *Synechococcaceae* for its high content of methionine and threonine. Furthermore, *Synechococcaceae* could be offered as a natural additive for the pigmentation of egg yolk and broiler meat due to its darker shade of yellowness. It is important to emphasize that the nutritional value and production cost of microalgae must be considered to choose the right one as an alternative feedstuff.

It is important to note that the determination of digestibility and supplementation amount for poultry nutrition causes some challenges in practice when used as a feed material or feed additive. Furthermore, when regarding some difficulties in microalgae production, for example, high production cost, downstream processing, and storage conditions, the economic feasibility should be worked on in detail to produce large amounts of microalgae feedstock that might be potentially produced more cheaply than existing feedstuffs. This is the critical point for microalgae production, and the sectoral demand for microalgae biomass is growing. Therefore, in the future, more research is needed about cultivation strategies, for the identification of the sustainable and economical production of biomass.

## Figures and Tables

**Figure 1 animals-13-03431-f001:**
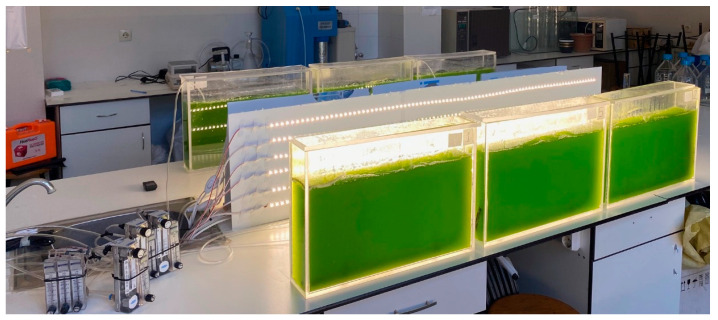
Algae production in flat panel PBRs.

**Figure 2 animals-13-03431-f002:**
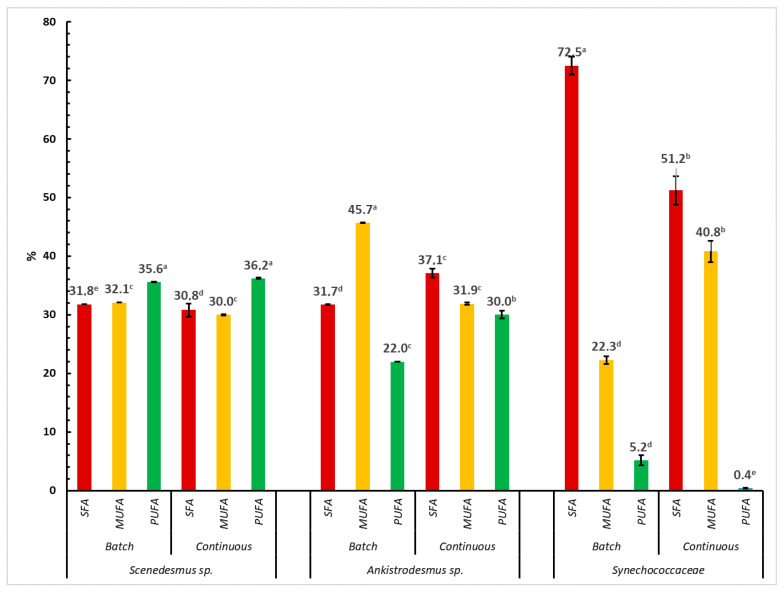
Comparison of saturated and unsaturated fatty acids from different microalgae species (*Scenedesmus* sp., *Ankistrodesmus* sp., *Synechococcaceae*) and cultivation practice. ^a–e^ Differences in letters shows the differences among the experimental groups.

**Figure 3 animals-13-03431-f003:**
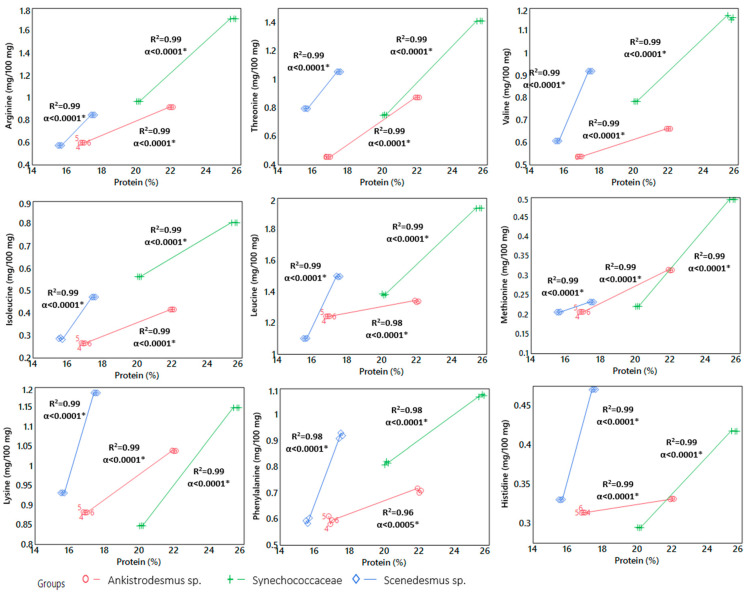
Correlations between protein content and amino acid content of algae biomass. (* shows the significant differences among the experimental groups).

**Figure 4 animals-13-03431-f004:**
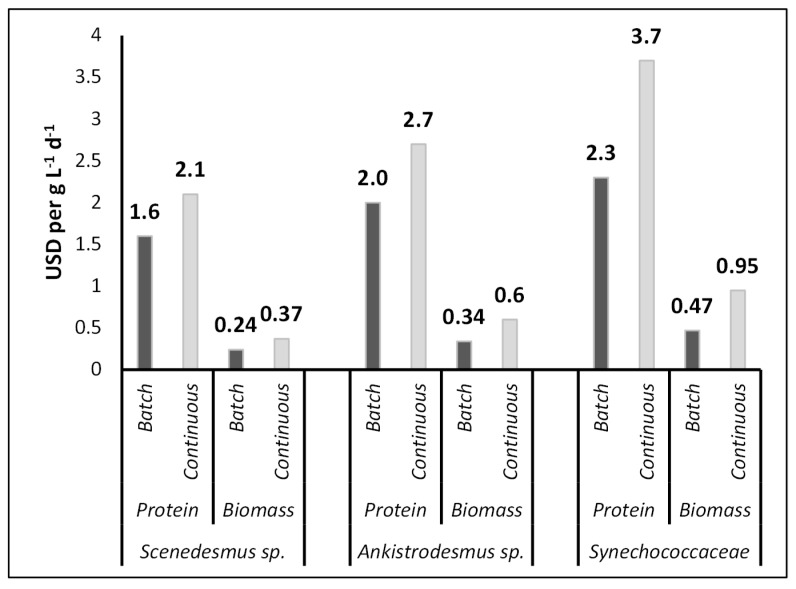
Algal biomass and protein production costs under batch and continuous cultivation modes for *Scenedesmus* sp., *Ankistrodesmus* sp., and *Synechococcaceae*.

**Table 1 animals-13-03431-t001:** Growth conditions for microalgae cultures.

Condition Items	Treatment PBRs (Batch and Continuous Culture)	Inoculum PBRs (Batch Culture)
Photobioreactor	The flat-plate reactor, 10 L	The flat-plate reactor, 10 L
Temperature (°C)	24 ± 2	24 ± 2
pH	7.0 ± 0.2	7.0 ± 0.2
Medium	Bold Basal Medium	Bold Basal Medium
CO_2_ supply	1200 ppm	Air
Light intensity	180–200 µmol m^−2^ s^−1^	60–80 µmol m^−2^ s^−1^
Cultivation period	7 days	7 days

**Table 2 animals-13-03431-t002:** Biomass production and biochemical composition of *Scendesmus* sp., *Ankistodesmus* sp., and *Synechococcaceae* under batch and continuous operations at 200 μmol m^−2^ s^−1^; CO_2_ feeding concentration = 0.15%; air flow rate = 0.5 vvm.

Algae/ Cultivation Mode	*Scendesmus* sp.		*Ankistodesmus* sp.		*Synechococcaceae*		SEM	*p*-Value
	Batch	Continuous	Batch	Continuous	Batch	Continuous		
**Dry biomass** **concentration (g L^−1^)**	0.97 ± 0.04 ^a^	0.57 ± 0.22 ^c^	0.76 ± 0.03 ^b^	0.58 ± 0.02 ^bc^	0.3 ± 0.1 ^d^	0.25 ± 0.06 ^d^	0.03	0.0001 *
**Dry biomass** **productivity (mg L^−1^ d^−1^)**	230.7 ± 10.6 ^a^	180.9 ± 22.4 ^ab^	164.2 ± 7.1 ^ab^	114.2 ± 5.5 ^bc^	119.0 ± 35.9 ^bc^	71.4 ± 28.6 ^c^	15.1	0.0131 *
**Carbohydrate** **content (%)**	50.1 ± 3.0 ^b^	57.2 ± 1.9 ^a^	17.2 ± 1.7 ^c^	12.1 ± 0.3 ^d^	7.9 ± 0.3 ^e^	6.3 ± 0.1 ^e^	0.53	0.0001 *
**Carbohydrate** **productivity (mg L^−1^ d^−1^)**	115.7 ± 9.8 ^a^	104.0 ± 5.9 ^a^	28.2 ± 2.0 ^b^	13.8 ± 0.23 ^b^	9.2 ± 1.6 ^b^	4.5 ± 1.8 ^b^	3.1	0.0002 *
**Protein** **content (%)**	15.6 ± 0.9 ^e^	17.5 ± 0.5 ^d^	16.9 ± 0.3 ^d^	22.0 ± 0.1 ^b^	20.4 ± 0.1 ^c^	25.6 ± 0.4 ^a^	0.24	0.0001 *
**Protein** **Productivity (mgL^−1^ d^−1^)**	36.0 ± 3.2 ^a^	31.9 ± 1.8 ^ab^	27.7 ± 1.4 ^ab^	25.1 ± 1.1 ^ab^	24.1 ± 1.5 ^ab^	18.2 ± 1.7 ^b^	2.87	0.0001 *
**Lipid** **content (%)**	31.7 ± 0.1 ^d^	30.8 ± 0.4 ^e^	31.8 ± 1.1 ^d^	37.1 ± 0.8 ^c^	72.4 ± 1.5 ^a^	51.2 ± 1.3 ^b^	0.03	0.0001 *
**Lipid** **productivity (mg L^−1^ d^−1^)**	73.1 ± 8.6 ^ab^	55.7 ± 2.3 ^abc^	52.2 ± 1.8 ^bc^	42.4 ± 1.9 ^bc^	86.2 ± 8.4 ^a^	36.5 ± 6.7 ^c^	5.83	0.0365 *

^a–e^ Differences in letters within rows indicate significant differences among the experimental groups. * shows the significant differences among the experimental groups.

**Table 3 animals-13-03431-t003:** Essential amino acid composition of algae biomass harvested from different microalgae species (*Scenedesmus* sp., *Ankistrodesmus*, and *Synechococcaceae*) and cultivation practices.

Main Effects	Arginine	Threonine	Valine	Isoleucine	Leucine	Methionine	Lysine	Phenylalanine	Histidine
**Algae species**									
*Scenedesmus* sp. *(A1)*	0.71 ^c^	0.92 ^b^	0.76 ^b^	0.38 ^b^	1.30 ^b^	0.22 ^c^	1.06 ^a^	0.76 ^b^	0.40 ^a^
*Ankistrodesmus* sp. *(A2)*	0.76 ^b^	0.66 ^c^	0.60 ^c^	0.34 ^c^	1.29 ^c^	0.26 ^b^	0.96 ^c^	0.65 ^c^	0.32 ^c^
*Synechococcaceae (A3)*	1.34 ^a^	1.08 ^a^	0.97 ^a^	0.68 ^a^	1.66 ^a^	0.36 ^a^	0.99 ^b^	0.94 ^a^	0.36 ^b^
SEM	0.0003	0.0004	0.002	0.0007	0.001	0.0002	0.0002	0.004	0.0001
**Cultivation**									
Batch	0.71 ^b^	0.67 ^b^	0.64 ^b^	0.37 ^b^	1.24 ^b^	0.21 ^b^	0.89 ^b^	0.67 ^b^	0.31 ^b^
Continuous	1.16 ^a^	1.11 ^a^	0.91 ^a^	0.56 ^a^	1.59 ^a^	0.35 ^a^	1.12 ^a^	0.90 ^a^	0.41 ^a^
SEM	0.0002	0.0004	0.001	0.0006	0.001	0.0002	0.0002	0.003	0.0001
**Algae species** **× Cultivation modes**									
*A1* × Batch	0.57 ^f^	0.79 ^d^	0.60 ^e^	0.29 ^e^	1.10 ^f^	0.21 ^e^	0.93 ^d^	0.59 ^e^	0.33 ^c^
*A2* × Batch	0.60 ^e^	0.45 ^f^	0.54 ^f^	0.27 ^f^	1.24 ^e^	0.21 ^e^	0.88 ^e^	0.60 ^e^	0.31 ^d^
*A3* × Batch	0.97 ^b^	0.75 ^e^	0.78 ^c^	0.56 ^b^	1.38 ^c^	0.22 ^d^	0.85 ^f^	0.82 ^c^	0.29 ^e^
*A1* × Continuous	0.85 ^d^	1.05 ^b^	0.92 ^b^	0.47 ^c^	1.50 ^b^	0.23 ^c^	1.19 ^a^	0.92 ^b^	0.47 ^a^
*A2* × Continuous	0.91 ^c^	0.87 ^c^	0.66 ^d^	0.42 ^d^	1.34 ^d^	0.31 ^b^	1.04 ^c^	0.71 ^d^	0.33 ^c^
*A3* × Continuous	1.71 ^a^	1.41 ^a^	1.16 ^a^	0.81 ^a^	1.94 ^a^	0.50 ^a^	1.15 ^b^	1.07 ^a^	0.42 ^b^
SEM	0.0003	0.0007	0.002	0.001	0.002	0.0002	0.0003	0.006	0.0002
***p*-Values**									
Algae species	*0.0001*	*0.0001*	*0.0001*	*0.0001*	*0.0001*	*0.0001*	*0.0001*	*0.0001*	*0.0001*
Cultivation modes	*0.0001*	*0.0001*	*0.0001*	*0.0001*	*0.0001*	*0.0001*	*0.0001*	*0.0001*	*0.0001*
Algae × Cultivation	*0.0001*	*0.0001*	*0.0001*	*0.0001*	*0.0001*	*0.0001*	*0.0001*	*0.0001*	*0.0001*

^a–f^ Differences in letters within columns indicate significant differences among the experimental groups.

**Table 4 animals-13-03431-t004:** Non-essential amino acid composition of algae biomass harvested from different microalgae species (*Scenedesmus* sp., *Ankistrodesmus*, and *Synechococcaceae*) and cultivation practices.

Main Effects	Serine	Alanine	Proline	Tyrosine	Aspartic Acid	Ornithine	Glutamic Acid	Glycine
**Algae species**								
*Scenedesmus* sp. *(A1)*	0.61 ^b^	1.58 ^b^	1.09 ^a^	0.37 ^c^	1.84 ^b^	0.11 ^c^	2.07 ^b^	1.22 ^b^
*Ankistrodesmus* sp. *(A2)*	0.26 ^c^	1.46 ^c^	0.94 ^c^	0.44 ^b^	1.07 ^c^	0.13 ^a^	1.50 ^c^	0.57 ^c^
*Synechococcaceae (A3)*	0.74 ^a^	2.03 ^a^	0.99 ^b^	0.88 ^a^	1.99 ^a^	0.12 ^b^	2.78 ^a^	1.39 ^a^
SEM	0.0001	0.001	0.00004	0.0002	0.001	0.0001	0.001	0.001
**Cultivation**								
Batch	0.38 ^b^	1.53 ^b^	0.96 ^b^	0.48 ^b^	1.45 ^b^	0.123 ^b^	1.93 ^b^	1.22 ^a^
Continuous	0.69 ^a^	1.85 ^a^	1.06^a^	0.65 ^a^	1.81 ^a^	0.126 ^a^	2.30 ^a^	0.90 ^b^
SEM	0.0001	0.001	0.00004	0.0002	0.0009	0.0001	0.0009	0.001
**Algae species** **× Cultivation modes**								
*A1* × Batch	0.36 ^d^	1.41 ^f^	1.00 ^c^	0.33 ^f^	1.42 ^d^	0.116 ^e^	1.73 ^d^	1.15 ^d^
*A2* × Batch	0.26 ^e^	1.50 ^d^	0.93 ^f^	0.38 ^e^	1.01 ^f^	0.133 ^a^	1.49 ^f^	1.13 ^e^
*A3* × Batch	0.52 ^c^	1.68 ^c^	0.94 ^e^	0.74 ^b^	1.91 ^c^	0.121 ^d^	2.57 ^b^	1.37 ^b^
*A1* × Continuous	0.86 ^b^	1.76 ^b^	1.17 ^a^	0.40 ^d^	2.25 ^a^	0.121 ^d^	2.42 ^c^	1.29 ^c^
*A2* × Continuous	0.26 ^f^	1.42 ^e^	0.94 ^d^	0.51 ^c^	1.12 ^e^	0.132 ^b^	1.51 ^e^	0.001 ^f^
*A3* × Continuous	0.96 ^a^	2.38 ^a^	1.05 ^b^	1.02 ^a^	2.06 ^b^	0.123 ^c^	2.98 ^a^	1.41 ^a^
SEM	0.0001	0.001	0.00007	0.0003	0.002	0.0002	0.002	0.001
***p*-Values**								
Algae	0.0001	0.0001	0.0001	0.0001	0.0001	0.0001	*0.0001*	*0.0001*
Culture	0.0001	0.0001	0.0001	0.0001	0.0001	0.0001	*0.0001*	*0.0001*
Algae × Culture	0.0001	0.0001	0.0001	0.0001	0.0001	0.0001	*0.0001*	*0.0001*

^a–f^ Differences in letters within columns indicate significant differences among the experimental groups.

**Table 5 animals-13-03431-t005:** Color characteristics of algae biomass harvested from different microalgae species (*Scenedesmus* sp., *Ankistrodesmus*, and *Synechococcaceae*) and cultivation practices.

Main Effects	L* Value	a* Value	b* Value	C*_ab_ Value	α° Value
**Algae species**					
*Scenedesmus* sp.	17.8 ^b^	−4.8	17.7 ^a^	18.2 ^a^	105.2 ^b^
*Ankistrodesmus* sp.	27.3 ^a^	−4.2	19.8 ^a^	20.3 ^a^	102.2 ^c^
*Synechococcaceae*	23.7 ^a^	−4.6	5.4 ^b^	7.3 ^b^	120.2 ^a^
SEM	1.2	0.28	1.1	1.1	0.5
**Cultivation**					
Batch	22.4	−3.1 ^a^	13.8	14.1	103.3 ^b^
Continuous	23.3	−5.9 ^b^	14.8	16.4	115.1 ^a^
SEM	0.9	0.2	0.9	0.9	0.4
**Algae species × Cultivation modes**					
*Scenedesmus* sp. × Batch	21.7 ^bc^	−4.6 ^b^	17.6 ^a^	17.9 ^a^	104.8 ^bc^
*Ankistrodesmus* sp. × Batch	24.7 ^bc^	−4.1 ^b^	20.9 ^a^	21.4 ^a^	101.1 ^d^
*Synechococcaceae* × Batch	21 ^c^	−0.7 ^a^	2.9 ^c^	2.9 ^c^	103.9 ^bc^
*Scenedesmus* sp. × Continuous	13.9 ^d^	−4.9 ^b^	17.8 ^a^	18.5 ^a^	105.6 ^b^
*Ankistrodesmus* sp. × Continuous	29.6 ^a^	−4.3 ^b^	18.7 ^a^	19.2 ^a^	103.2 ^cd^
*Synechococcaceae* × Continuous	26.4 ^ab^	−8.4 ^c^	7.9 ^b^	11.6 ^b^	136.5 ^a^
SEM	1.7	0.39	1.5	1.6	0.8
***p*-Values**					
Algae species	<0.0001	0.38	<0.0001	<0.0001	<0.0001
Culture	0.5472	<0.0001	0.44	0.087	<0.0001
Algae × Culture	<0.0003	<0.0001	0.06	<0.0038	<0.0001

^a–d^ Differences in letters within columns indicate significant differences among the experimental group.

**Table 6 animals-13-03431-t006:** Economic estimates of PBR system.

Economic Estimates of PBR System	
**Capital Cost**	**$519.24/PBR**
Flowmeter_$346.7 (air_$45.7 + CO_2__$301)	
Air Pump (1.5 Hp 50 Lt–1.1 kW)_$96	
Lighting_$14.68	
Daylight Leds (14.4 W/m)_$3.5/meter	
Acrylic Sheets (0.47 m^2^/each PBR)_$52.8/PBR	
Bubble Wall_$5.56	
**Batch Cultivation Mode Operating Cost**	**$0.056 L^−1^ day^−1^**
Nutrients_$0.02/Lday	
Aeration Energy Consumption_$0.034/Lday	
Lighting Energy_$0.0024/Lday	
**Continuous Cultivation Mode Operating Cost**	**$0.068 L^−1^ day^−1^**
Nutrients_$0.032/Lday	
Aeration Energy Consumption_$0.034/Lday	
Lighting Energy_$0.0024/Lday	

## Data Availability

All data sets collected and analyzed in the study are available from the corresponding author upon reasonable request.

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
