# Peer review of "Nutritional Value of Microalgae and Cyanobacteria Produced with Batch and Continuous Cultivation: Potential Use as Feed Material in Poultry Nutrition"

_animals, 2023, doi:10.3390/ani13213431_

Round 1

Reviewer 1 Report

Comments and Suggestions for Authors

The work aims to assess microalgae usage as an alternative to dietary protein source particularly in poultry industry. In that regards three microalgal species were grown in batch mode and continues mode to enhance the biomass productivity and nutritional values utilising BB media under standard operating conditions in 10 L flat plate PBRs. The results difference in fatty acid, protein and carbohydrates compositions were depicted in results.

A major revision is needed for the manuscript in order to elucidate the underlying causes of the color characteristics resulting from microalgal supplementation and its correlation with the pigmentation of egg yolk and meat. The listed comments can be taken into account during this revision process.

1.      Keywords: Microalgae, poultry and amino acids are redundant. The author can use this space for complimentary words

2.      Synechococcacea is a cyanobacterial strain, title can be modified accordingly.

3.      Abstract- Instead of using the term different, it is suggested to specify as three microalgae species.

4.      Abstract-The research results are not clearly reflected in the abstract. Novelty of the research is not emphasized. It should clearly state the essence of problem, what was done to address it and what are the novel results and ultimate recommendation.

5.      Line 36: what does the author mean by free range and organic production.

6.      Section 2.2, Table 1: Why is the light intensity is varying in control and treatment PBRs?

7.      The air sparged in control condition will definitely be having certain ratio of CO2, that can also be considered?

8.      How was the lipid analysis performed? Biomass in terms of dry cell weight has to be indicated in the manuscript.

9.      Line 111, specify the unit of duration rate.

10.  Does stress phase or nutritional deficit implement during the cultivation of microalgae for lipid and protein accumulation? If not, could you provide insights into the metabolic mechanisms underlying by Scenedesmus sp., Ankistrodesmus sp., and Synechococcaceae.

11.  Results and Discussion-The discussion for obtained results needs to be critical and scientific evidences supporting the results needs to be included to improve the scientific aspect.

12.  What is the biomass productivity in all the treatments?

13.  Table 2: a,b,c,d, etc signifies what? Please specify

14.  Represent each scientific names within italics throughout the manuscript.

15.  Lipid composition is not clearly discussed but the author states docosahexaenoic acid (DHA) stimulates growth and body weight. Rephrase the statement.

16.  How does the colour characteristics of microalgae species corelate with pigmentation of egg yolk and meat? Does the author perform any supplementation experiments?

17.  Figure 3: What is the output from corelation studies of protein and amino acid profiling?

18.  The aim of using algal biomass for poultry ration can’t be economical while utilising BBM, how does authors defend the work with conventional poultry farming.

19.  Any recommendation for future work in this regard?

20.  Economic evaluation of the process needs to be briefly elaborated. The cost analysis of each cultivation process with respect to cultivation conditions needs to be addressed.

Comments on the Quality of English Language

Please see report

Reviewer 2 Report

Comments and Suggestions for Authors

In the present study, the authors aim to investigate the usage of algae as a feed ingredient in poultry nutrition. Three kinds of microalgae (Scenedesmus sp., Ankistrodesmus sp., and Synechococcaceae) and two cultivation methods (batch and continuous cultivation) were actually applied, and the lipid, protein, carbohydrate, amino acid profiles, and color characteristics were evaluated. This work has certainly a great potential to be interesting for a general audience. However, I recognize several concerns to be clarified as well as issues for improvement.

Major comments:

1. Based on the results, it is undeniable that the contents of nutritional ingredients in microalgae are similar to the traditional used feed ingredients. However, no animal experiments were conducted in this study. Therefore, the authors can not directly obtain that daily used microalgae has benefits in animal health, growth performance, and product quality. We suggest a actual animal experiments is needed. 

2. Although lipid, protein, carbohydrate, amino acid, and color characteristics of three microalgae was detected, we can not confirm the safety of these products. The safety related indexes should be detected, such as heavy metals, pesticide residues, and pathogenic bacterium.

3. In this manuscript, the deeper mechanism related to amino acid generation is limited. For example, the continuous cultivation resulted in a higher level of essential amino acids, except glycine. This phenomenon should be comprehensively analyzed to find out the essential reasons, rather than staying on the surface.

Specific comments:

1. There is an extra dot in the Scenedesmus sp.. in line 12.

2. The words of microalgae are not italicized, which are on lines 73-74, 191, 219-220, 233, 234, 271, 292-293 and 344 respectively. Please keep them in a uniform format in the text.

3. The ℃ after "210" in line 131 and "110" in line 147 is wrong. The ℃ format of the manuscript is not unified.

4. Is the identification of FAME in the sample determined only by retention times? It seems that it is not rigorous.

5. The word “acid” in line 281 should be specific to avoid ambiguity.

6. The sentences in lines 310-313 are confusing.

7. There is an error in the letters within columns, such as “Cultivation - Threonine” and “Algae species × Cultivation modes - Histidine”.

8. The standard deviation should be added to the data of parallel samples, such as Table 2 to Table 5 and Figure 2, which need to be supplemented.

Comments on the Quality of English Language

Minor editing of English language required

Reviewer 3 Report

Comments and Suggestions for Authors

The authors investigated the effect of growth condition (batch or continuous) on the nutritional accumulation of three micro algae and they suggest that the micro algae can be utilized as feed ingredients for poultry. This approach can suggest an eco-friendly farming way, so that is valuable for industrial field.

However, this MS prioritizes the research requirement rather than the interpretation of the data. Of course the chemical composition is very important but the describing for the variation according to the growth condition is very weak.

Major comments

1. It is necessary to provide the reasons for choosing the micro algae and comparing the two culture condition.

2. Why did the CO2 supply and light intensity differ in the two culture condition?

3. The biomass by the two culture condition should be provided and compared.

4. The chemical composition varies on the algal species. Nevertheless, why did the authors compare the cultivation modes in all the Tables, it confuses the results.

5. Please reflect the research purpose and results in the conclusion!

Minor comments

1. Please check the italic notation of the algal species

2. Line 118, ‘chemical content’, revise by ‘composition’.

3. Line 264, ‘Significant interactions were…’, revise by ‘Significant variations were…’

4. Line 265, ‘The SFA content’, revise by ‘The percent ratio of SFA in the total fatty acids’

Comments on the Quality of English Language

Minor editing of English language required

Round 2

Reviewer 2 Report

Comments and Suggestions for Authors

Accept in present form

Reviewer 3 Report

Comments and Suggestions for Authors

The manuscript has been appropriately revised in accordance with reviewer's comments.

Comments on the Quality of English Language

A minor spelling check is required.